# Identification of Spatial Specific Lipid Metabolic Signatures in Long-Standing Diabetic Kidney Disease

**DOI:** 10.3390/metabo14110641

**Published:** 2024-11-20

**Authors:** Yiran Zhang, Hai-Long Piao, Di Chen

**Affiliations:** 1Key Laboratory of Phytochemistry and Natural Medicines, Dalian Institute of Chemical Physics, Chinese Academy of Sciences, Dalian 116023, China; zhangyiran@dicp.ac.cn; 2University of Chinese Academy of Sciences, Beijing 100049, China

**Keywords:** diabetic kidney disease, injured thick ascending limb (iTAL), injured proximal tubule (iPT), spatial metabolomics, single-cell RNA sequencing

## Abstract

**Background:** Diabetic kidney disease (DKD) is a major complication of diabetes leading to kidney failure. **Methods:** This study investigates lipid metabolism profiles of long-standing DKD (LDKD, diabetes duration > 10 years) by integrative analysis of available single-cell RNA sequencing and spatial multi-omics data (focusing on spatial continuity samples) from the Kidney Precision Medicine Project. **Results:** Two injured cell types, an injured thick ascending limb (iTAL) and an injured proximal tubule (iPT), were identified and significantly elevated in LDKD samples. Both iTAL and iPT exhibit increased lipid metabolic and biosynthetic activities and decreased lipid and fatty acid oxidative processes compared to TAL/PT cells. Notably, compared to PT, iPT shows significant upregulation of specific injury and fibrosis-related genes, including *FSHR* and *BMP7*. Meanwhile, comparing iTAL to TAL, inflammatory-related genes such as *ANXA3* and *IGFBP2* are significantly upregulated. Furthermore, spatial metabolomics analysis reveals regionally distributed clusters in the kidney and notably differentially expressed lipid metabolites, such as triglycerides, glycerophospholipids, and sphingolipids, particularly pronounced in the inner medullary regions. **Conclusions:** These findings provide an integrative description of the lipid metabolism landscape in LDKD, highlighting injury-associated cellular processes and potential molecular mechanisms.

## 1. Introduction

Diabetic kidney disease (DKD) is a predominant cause of end-stage renal disease (ESRD) [1,2,3], and its prevalence closely linked to lifestyle and dietary habits [4]. As the primary microvascular complication arising from diabetes, DKD is frequently associated with dysregulated lipid metabolism, a critical factor in the development of cirrhosis and metabolic syndrome [5]. Historically, research on kidney disease has focused on the glomerular and tubular components of the nephron, recognizing their vital roles in kidney function [6,7]. The fatty acid oxidation pathway has also been a subject of interest due to its essential contribution to kidney health.

Despite elucidating connections between specific metabolic pathways and cellular lipid classes, current studies lack a holistic perspective on disease progression, especially regarding the intricate relationship between lipid metabolism and the pathogenesis of DKD, particularly in its fibrotic progression. This complexity may prompt a reorientation in research priorities. Recent studies have expanded the investigation to encompass various lipid species, such as sterols, phospholipids, and sphingomyelin, reflecting an increased understanding of the multifactorial pathogenesis of DKD [8]. Despite these advances, our knowledge of DKD fibrosis is still incomplete, especially concerning the metabolic signatures at spatial levels. 

While traditional bulk omics approaches have yielded valuable insights, limitations in capturing the spatial and cellular heterogeneity are still present in kidney fibrosis. The emergence of single-cell transcriptomics, coupled with spatial transcriptomics and metabolomics, offers an unprecedented opportunity to dissect the intricate interplay between cellular populations and their microenvironment during the fibrotic process. 

In this study, we investigated the cellular and spatial molecular changes regarding lipid metabolism in DKD by integrative analyses of single-cell transcriptomics (*n* = 41), spatial transcriptomics (*n* = 16), and spatial metabolomics (*n* = 10) data from the Kidney Precision Medicine Project (KPMP) cohort [9]. This allowed us to conduct an in-depth analysis of the metabolic differences and characteristics between patients with long-standing DKD (LDKD, diabetes duration > 10 years) and healthy donors across multiple dimensions.

Our primary objective is to delineate the metabolic signatures associated with advanced DKD, thereby providing insights into the DKD process. This comprehensive analysis aims to provide new insights regarding lipid metabolism in LDKD.

## 2. Materials and Methods

### 2.1. Participant Selection and Data Acquisition

Our study was mainly based on the multi-omics data from the Kidney Precision Medicine Project (KPMP) (https://www.kpmp.org/, accessed on 7 October 2023), which encompassed spatial transcriptomics, spatial metabolomics, and single-cell transcriptomics data. The spatial metabolomics data were downloaded from METASPACE [10] based on the links provided by KPMP. We independently selected subjects for analysis, including both patients with LDKD and healthy donors.

To select the samples for the study, we first screened healthy donors and chronic kidney disease (CKD) patients. To elucidate the metabolic characteristics of LDKD patients with prolonged diabetes duration, we focused on individuals with over a decade of diabetes history. For the spatial multi-omics sliced samples, which additionally involved the spatial distributions, we removed samples with spatial discontinuities. A total of 37 LDKD patients and 26 healthy donors were included in the study. The scRNA-seq data included samples from 25 LDKD patients and 16 healthy donors. The spatial transcriptomics data included samples from 9 LDKD and 7 healthy donors. The spatial metabolomics data included samples from 7 LDKD and 3 healthy donors. The detailed information for the selected samples is summarized in Appendix A.

### 2.2. Single-Cell RNA Sequencing (scRNA-seq) Data Analysis

Preprocessing of the dataset was performed using Seurat [11] (v5.0.1) package to exclude low-quality cells (nFeature_RNA < 10,000 & nFeature_RNA > 500 & percent.mito < 50). Then, following the standard data processing workflow, functions such as NormalizeData, FindVariableFeatures, ScaleData, and RunPCA were applied. The RunHarmony function was executed using the method in Harmony [12] (v1.2.0) to remove batch effects. To ensure the reproducibility of the results, a random seed was set to 42. Subsequent analyses utilized the Harmony results, with the resolution set to 0.3 during clustering. Dimensionality reduction was achieved using UMAP, and individual cell types were annotated based on the expression of lineage-specific markers reported in previous literature [13,14,15,16,17,18,19].

### 2.3. Pathway Enrichment Analysis 

Pathway enrichment analysis was conducted using the clusterProfiler R package [20] (v4.7.1.003). Upregulated genes in each cell type were used as the input gene list, with pathway information sourced from the Kyoto Encyclopedia of Genes and Genomes (KEGG) and GO Biological Process (GO BP) databases. The upregulated genes identified in RNA analysis had a minimum expression percentage set to 0, with a significance threshold for gene expression changes set at 0.1.

### 2.4. Pathway Activity Profiling

Pathway activity in each cell was determined based on the AUCell R package [21] (v1.21.2), which employed gene set enrichment analysis to assess and score pathways at the cellular level. Subsequently, the AUC scores, which served as an indicator of pathway activity, were mapped to the UMAP embedding using the ggplot2 R package [22] (v3.5.0). The GO BP lipid-associated pathways were identified based on pathway names containing terminology related to fatty acids, lipids, cholesterol, and steroids. These maps were refined using a hot-spot removal step to better represent spatial differences in expression levels, employing the method of replacing the top 1% of expression values with the 99th percentile expression value, while keeping all other values unchanged.

Specific cluster comparisons, such as iPT versus PT and iTAL versus TAL, were performed within the GO Biological Process (GO BP) lipid-associated pathways. *p*-values were determined using the Wilcoxon test and subsequently adjusted using the Benjamini–Hochberg (BH) method. Significance is annotated as follows: ns (not significant) for *p* > 0.05, * for *p* ≤ 0.05, ** for *p* ≤ 0.01, *** for *p* ≤ 0.001, **** for *p* ≤ 0.0001. In these four comparisons, only pathways with −log10 adjusted *p*-values (−log10 (p.adj)) above the mean −log10 (p.adj) value across all pathways were presented.

Injured-cluster comparisons in spatial transcriptome data were analogous to those in scRNA data. However, to ensure cluster singularity, only cells with a final proportion exceeding 50% of the cell population were considered for inclusion.

### 2.5. Cell–Cell Interaction Inference

To elucidate metabolic crosstalk and identify significant interactions, specifically ligand–receptor (L-R) pairs, in either LDKD or healthy samples, we utilized the CellChat R package [23] (v2.1.2). The package enables the inference of potential signaling interactions between cells by referencing a predefined database that catalogs ligand–receptor pairs. Significant ligand–receptor pairs were identified through permutation testing. To examine metabolic communications, we specifically selected the “Non-protein Signaling” subset of this database. Especially, in the analysis of spatial transcriptomes, only cells with high purity, where the predominant cell type constitutes more than 50%, were considered for the study.

### 2.6. Spatial Transcriptomics Data Analysis

We utilized Seurat to analyze the spatial transcriptomics data from KPMP, applying a pre-processing threshold of (nFeature_RNA > 500 & percent.mito < 50). The Seurat standardization process was followed using NormalizeData and ScaleData functions, and batch effect correction was conducted using the RunHarmony function. The UMAP dimensionality reduction was applied to the data processed by the Harmony algorithm, using the first 30 dimensions. The clustering was performed with a resolution of 0.5.

### 2.7. Deconvolution of Spatial Transcriptomics Cell Types Based on scRNA-seq Data

The determination of spatial cluster names was aligned with the cell annotations established in single-cell RNA sequencing (scRNA) analysis using the deconvolution method (function name FindTransferAnchors and TransferData, normalization method is “LogNormalize”) from the Seurat package. The RNA assay of the scRNA-seq and the RNA assay of the spatial transcriptomics data were respectively utilized as the input reference and query objects. This deconvolution method can predict the potential proportion of each cell type in each spatial spot based on the cell type labels of the scRNA-seq data. For each data point, we designated the most abundant cell type as the representative cell type of that point.

### 2.8. Composition Changes in Spatial Cell Types 

We first identified individual spatial cell types and ranked the points of each specific cell type by their proportion. Subsequently, the proportions of other cell types can be displayed. The trend line was smoothed using the “Locally Estimated Scatterplot Smoothing” (loess) method, with the underlying model formula expressed as y∼x.

### 2.9. Spatial Metabolomics Data Analysis

The molecular annotations and pixel intensity matrices for all samples at each sampling point in the spatial metabolomics experiment were obtained from METASPACE. Initially, each sample was normalized using the root mean square (RMS) method. Subsequently, to mitigate potential batch effects in subsequent analyses, all samples underwent batch correction using the “RunHarmony” method in the Seurat package. The subsequent data processing steps involve identifying highly variable features, standardizing the data, dimensionality reduction, constructing the nearest neighbor network, and determination of cluster numbers. These functions were all derived from the Seurat v5.0.1 package, specifically including FindVariableFeatures, ScaleData, RunPCA, RunUMAP, FindNeighbors, and FindClusters. Ultimately, 12 clusters were identified with a resolution set at 0.1. Cluster-specific metabolites were obtained through FindAllMarkers and confirmed via spatial distribution maps of metabolites. These maps were refined using a hot-spot removal step to better represent spatial differences in expression levels, employing the method of replacing the top 1% of expression values with the 99th percentile expression value, while keeping all other values unchanged.

### 2.10. Analysis of Differential Metabolite Class Composition

For analysis of lipid-related metabolite composition, we calculated differentially expressed metabolites in LDKD and healthy participants using the Wilcoxon test and selected metabolites with a *p*-value adjustment of less than 0.05. The categorization information of lipid-related metabolites was obtained from the KEGG compound database and categorized by merging the lipid category information from br08001 and br08002.

### 2.11. Spatial Distribution Similarity of Lipids and MCs

We first collected expression data for the lipid metabolites of interest and the characteristic metabolites of the metabolic clusters within the samples. Differential metabolites of each metabolic cluster with an adjusted *p*-value less than 0.01 from the other clusters were selected, and the *p*-value adjustment method was FDR (False Discovery Rate). We kept only those sample points where both the target and the characteristic metabolites of the metabolic clusters were expressed. The similarity between each pair of metabolites was calculated using the “spearman” with the cor.test function from the stats R package [24].

## 3. Results

### 3.1. Analysis of Kidney Cellular Composition at Single-Cell Level

We analyzed multi-omics data from the Kidney Precision Medicine Project (KPMP) cohort [9], which included 37 patients with long-standing diabetic kidney disease (LDKD) and 26 healthy donors. LDKD patients had diabetes for more than 10 years (Figure 1a), and we summarized the clinical characteristics of the investigated patients and healthy donors (Figure 1a and Appendix A).

Next, we examined scRNA-seq data from 25 LDKD patients and 16 healthy donors. After applying quality filters and excluding mitochondrial and ribosomal genes, we obtained an average of 3249 cells per sample, enabling a detailed characterization of kidney cellular clusters in both conditions (Appendix A). Following the removal of batch effects (Appendix A) and dimensionality reduction with UMAP, we identified 12 distinct cell types through single-cell analysis, including thick ascending limb (TAL), injured TAL (iTAL), proximal tubule (PT), injured PT (iPT), distal convoluted duct cells (DCT), glomerular endothelial cells (EDC), principal cells of the collecting duct (CD-PC), intercalated cell from the collecting duct (CD-IC), mesangial cells (MES), macrophages (Mac), B cells (Bcell), and T cells (Tcell) (Figure 1b). 

As previously defined [13,14,15,16,17,18,19], the top marker genes for each cell type were presented (Figure 1c), with the most prominent gene in each cluster being emphasized (Figure 1d). Most of the cell types were identified in previous KPMP studies [9,25], but we redefined the specific cell types associated with injury traits, including the iTAL and iPT, which were previously described as adaptive TAL and adaptive PT. Our goal is to investigate the lipid metabolism profiles and spatial characteristics of these injured cell types in LDKD patients. The iTAL cell type was characterized by the *ITGB6* marker, which was associated with fibrosis [16]. Meanwhile, the iPT cell type was identified by the expression of *VCAM1* and was indicative of an injured or regenerative subpopulation of cells [17]. Both iTAL and iPT cell types demonstrated an enrichment of the *PROM1* gene. It is noteworthy that both *ITGB6* and *PROM1* have been implicated in injury-related processes [18]. Additionally, we observed significant increases in the relative abundance of immune cell clusters, encompassing T cells, macrophages, and B cells in LDKD patients compared to healthy donors, aligning with prior findings [15] (Appendix A). Meanwhile, the proportions of the iTAL and iPT clusters were markedly increased in LDKD samples (Figure 1e and Appendix A), which may be correlated with the extent of kidney damage.

To investigate the metabolic features of the identified clusters, we performed metabolic pathway enrichment analysis for each cluster (Appendix A). The results indicated that the PT cluster was predominantly linked to carbohydrate metabolism, energy metabolism, lipid metabolism, and amino acid metabolism (Appendix A). These metabolic functions were closely associated with the physiological role of the kidney proximal tubules, and in the context of kidney disease, the PT cells may experience oxidative stress. Cell types iPT, iTAL, and CD-PC exhibited significant enrichment in metabolic modules associated with “Glycan Biosynthesis”, with a particular emphasis on the biosynthesis of N-glycans and O-glycans (Appendix A). This highlights the pivotal role of glycosylation in adaptive responses to kidney injury [26,27] and in preserving its intricate physiological functions. Furthermore, PT, TAL, and CD-IC revealed significant fatty acid biosynthesis, degradation, and elongation. iPT, iTAL, and CD-PC demonstrated enrichment of sphingolipid metabolism and glycerophospholipid metabolism (Appendix A).

For a refined functional dissection of lipid metabolism across different clusters, we executed biological processes enrichment analysis based on the Gene Ontology (GO) database (Figure 2a). The TAL, DCT, and CD-IC cell types were predominantly characterized by enrichment in fatty acid degradation and oxidation processes, alongside glycerolipid biosynthesis and metabolic process (Figure 2a). The iTAL cell type was distinguished by its emphasis on lipid biosynthesis and metabolic processes, including membrane lipids, glycolipids, and sphingolipids (Figure 2a). Lipid translocation, especially phospholipid, is prominently exhibited in the iPT cluster (Figure 2a). MES manifests a close relation with steroids’ biological process, which was caused by inflammatory stress in LDKD [28]. Meanwhile, the immune cluster Mac demonstrated multifaceted regulation in lipid processes, encompassing lipid localization, storage, metabolism, transport, and their involvement as regulatory components of the kinase (Figure 2a). Collectively, each cell type signifies the importance of lipid metabolism in cellular function, energy production, membrane dynamics, and hormonal regulation.

### 3.2. Lipid Metabolism Dysregulated in LDKD Patients

Subsequently, we conducted pathway activity analysis and compared the pathway activity levels of different cell types in LDKD and healthy samples (Section 2.4). Initially, we focused on the sphingolipid metabolism, which exhibited significantly elevated activity in both the iPT and iTAL cell types (Figure 2b,d). Sphingolipids, integral components of cell and organelle membranes, played pivotal roles in signal transduction and energy generation. Disorder of the sphingolipid metabolic pathway could exert a profound influence on the kidney cell functional capabilities and structural stability [29,30].

Furthermore, the cholesterol metabolism also showed notable upregulation in both iPT and iTAL cell types in LDKD (Figure 2c,d). This upregulation may indicate heightened cholesterol accumulation within the glomeruli, potentially exacerbating endothelial damage and dysfunction [31]. When examining the cell–cell interactions regarding the cholesterol signaling pathway network (Section 2.5), notably, we found that the cell types Mac and iTAL served as senders in the healthy samples. However, in LDKD, only the self-interaction of iTAL and its interaction with iPT was presented (Figure 2e and Appendix A). Retinoic acid receptor-related orphan receptor A (*RORA*) participated in all significant ligand–receptor pairs within cholesterol signaling pathways (Appendix A), contributing to the prevention of kidney injury and fibrosis [32,33]. Meanwhile, 24-dehydrocholesterol reductase (*DHCR24*) is reported to be involved in the cholesterol biosynthesis process [34] and could act as an antioxidant [35]. It also possesses proapoptotic functions at the cellular level [36] and could potentially serve as a novel causal biomarker for the risk of type 2 diabetes [37]. The transition from healthy to LDKD involved a limited number of cellular interactions, with only a slight increase in the variety of participating ligands and receptors (Appendix A).

Next, we compared the activities of the iTAL/iPT cells to the corresponding TAL/PT cell types in either LDKD or healthy samples. The comparative assessment showed significant decreases in involvement in lipid and fatty acid metabolic and oxidative processes in the iTAL/iPT cell types compared to the TAL/PT cell types in LDKD patients (Appendix A). In contrast, there were increases in activities associated with sphingolipid and membrane lipid metabolism, as well as other lipid metabolic and biosynthetic processes (Appendix A). Moreover, iTAL clusters showed significant upregulations in pathways related to unsaturated fatty acid biosynthesis, cholesterol transport, and corticosteroid response in LDKD (Appendix A).

Moreover, we focused on membrane lipid processes, which were enriched by the iTAL and CD cells (Figure 2a). These processes exhibited generally increased activities in the iTAL and iPT cell types compared to the corresponding TAL/PT cell types in LDKD patients (Appendix A). Membrane lipids were crucial for cell structure and function, and they also played a role in signaling processes inside and outside the cells. In the context of DKD, disruptions in lipid metabolism can cause issues like insulin resistance and inflammation [38,39].

To elucidate the genes functional within the injured cluster, we scrutinized those exhibiting the most significant fold changes across lipids-associated pathways (Figure 2f). In the iPT cell type, a subset of upregulated pathway genes play a role in governing insulin secretion from pancreatic islet β-cells. Among these, the follicle-stimulating hormone receptor (*FSHR*) is implicated in the pathogenesis of postmenopausal diabetes [40] and kidney tubulointerstitial injury [41]. The ATP-binding cassette transporter A12 (*ABCA12*) is noted for its multifaceted role in the regulation of pancreatic lipid homeostasis and insulin secretion [42]. Additionally, certain genes showed a direct association with DKD. For instance, bone morphogenetic protein-7 (*BMP7*) exerts anti-fibrotic effects and modulates the epithelial–mesenchymal transition in DKD [43,44]. Claudin-1 (*CLDN1*), whose expression is suppressed by the kidney podocyte-specific transcription factor Sirt1, is implicated in the amelioration of diabetic albuminuria [45]. The iPT cell type downregulated gene, fatty acid-binding protein 1 (*FABP1*) is observed to increase in parallel with the progression of DKD [46].

In the iTAL cell type, the upregulated gene *ANXA3* exhibits different functions in various types of diseases and acts as a potent endogenous anti-inflammatory mediator in inflammation [47,48]. *IGFBP2* elevates inflammation and oxidative stress levels and promotes podocyte apoptosis, which are key pathological features of DKD [49].

### 3.3. Spatial Transcriptomics Reveals Insights into Altered Lipid Metabolism and Dysregulation

To characterize the transcriptional landscape of LDKD at the spatial level, we analyzed the spatial transcriptomics data of samples from LDKD patients and healthy donors, filtering low-quality data and correcting for batch effects (Appendix A, Section 2.6). Then, we estimated the proportions of different cell types in each spatial spot by applying the deconvolution method (Figure 3a and Appendix A, Section 2.7), confirming their characteristics through analysis of the primary contributing genes in each cell type (Figure 3b). For each spatial spot, we selected the cell type with the highest proportion as the representative cell type for further analysis. Based on this assumption, we observed the accompanying changes in the other cell types as the proportion of the injury-associated cell type iTAL or iPT increased (Figure 3c). Specifically, as the iTAL proportion increased, the iPT and DCT cell types significantly decreased (Figure 3c, left panel). Additionally, the MES cell type showed notable changes in its proportions, displaying a similar trend across the two injury-associated cell types (Figure 3c). The dynamics of these changes were particularly intriguing. As the iTAL proportion continues to rise, the TAL initially maintained its stability before undergoing a subsequent decline. In contrast, the cluster PT, influenced by an increase in the proportion of iPT, first experienced a rapid decline and then stabilized (Figure 3c). 

We further explored the genes showing significant changes between iTAL/iPT and TAL/PT. In LDKD, the genes commonly increased in the iTAL and iPT cell types are primarily related to injury and fibrosis (Figure 3d and Appendix A). For example, the immune-related gene *SERPINA3* is a biomarker involved in DKD kidney tubular injury [50]. *VCAM1* is associated with tubular injury and serves as a histological marker for poor DKD outcomes [51,52]. *ITGB6* is linked to fibrosis [16] and injury-related processes [18]. Additionally, LTF, an innate immune protein, exhibits anti-inflammatory properties [53]. However, several genes were oppositely expressed in the two injury-associated cell types, such as *GPX3* and *SLC12A3* (Figure 3d and Appendix A). *GPX3* reflects fat accumulation [54,55] and might serve as an early marker for kidney damage [56,57]. *SLC12A3* contributes to genetic susceptibility to DKD [58,59], and its polymorphisms are associated with end-stage renal disease [60]. We also highlighted the genes showing significant changes in each cell type (Appendix A). Among them, SERPINA3 frequently exhibited decreases in PT, MES, and B cell types, while showing an increase in the EDC cell type.

Furthermore, we investigated the spatially resolved metabolic characteristics associated with lipid pathways in iPT, revealing significant upregulation in steroid hormone signaling, glycolipid, glycerophospholipid, and other lipids metabolic and biosynthetic pathways compared to PT in LDKD patients (Figure 3e). In contrast, lipid and fatty acid-associated oxidation and catabolic processes were downregulated (Figure 3e). Additionally, in patients with LDKD, steroid hormone stimulus and response were significantly upregulated in iTAL compared to the TAL cluster (Figure 3e). The metabolic characteristic changes in the two injury-associated clusters closely resembled those observed at the single-cell level compared to their counterpart clusters (Appendix A). Among those pathways, the gene *SLC44A5*, which was increased in the iPT cell type (Figure 3f) and belongs to the choline transporter-like family SLC44 [61], also appeared in the scRNA-seq of iPT cells (Figure 2f). The decreased gene *HSD11B2* in the iPT cell type (Figure 3f), which is associated with intra-adipose cortisol levels and peripheral glucose uptake [62], may be essential for an optimized steroid therapy [63]. Taken together, these findings suggested that the metabolic reprogramming in iPT and iTAL cell types might play a crucial role in the pathogenesis of LDKD.

During our investigation into cell–cell interactions related to cholesterol signaling pathways, we observed that in the healthy state, the PT cell type uniquely served as a target for signals. In contrast, in LDKD samples, the iTAL, iPT, MES, and Mac cell types exclusively functioned as signal senders, with the TAL, PT, and CD cell types taking on dual roles (Figure 3g and Appendix A). In the LDKD state, the spatial interaction has expanded to include *RORC* as an additional receptor, in addition to *RORA* (Appendix A). *RORA* and *RORC* are part of the nuclear receptor family of intracellular transcription factors. Cholesterol is associated with the activation of *RORC*, which in turn is implicated in inflammatory and metabolic processes [64,65]. This highlights the increased cholesterol metabolic complexity in LDKD compared to healthy samples.

### 3.4. Spatial Metabolomics Elucidate Metabolic Regionalization in LDKD and Healthy Kidneys

To comprehensively understand the spatial metabolite distribution in LDKD, we conducted a clustering analysis of spatial metabolite imaging data from KPMP. The distribution of metabolite features denoted as nFeature_Metabolite was illustrated (Appendix A). Furthermore, after batch effect correction, dimensionality reduction, and clustering analysis, 12 distinct metabolic clusters (MCs), MC1 to MC12, were identified (Figure 4a,b and Appendix A). Next, we evaluated the proportions of these MCs in each slice (Figure 4c). We found that MC2, MC4, MC5, MC7, MC8, and MC9 were specific to LDKD patients (including those with proportions less than 0.1% in healthy donors), while clusters MC1, MC3, MC6, MC10, and MC11 exhibited markedly decreased levels in LDKD patients (Figure 4d). Additionally, for both LDKD and healthy samples, the spatial location of the metabolic clusters showed zonation distributions from the outside (e.g., MC1, MC2) to the inside (e.g., MC6, MC7) (Figure 4e and Appendix A), corresponding to the anatomical structure of the kidney from cortex to outer and inner medulla [66,67]. 

Distinct MCs were associated with corresponding zonation metabolites. For instance, clusters MC1 and MC2 were identified in the cortex region, with metabolites such as C_10_H_8_NO_4_, C_14_H_21_N_6_O_3_S, C_12_H_17_N_5_O_3_SCl, and C_17_H_19_N_2_O_5_S, enriched in specific clusters (Appendix A). Cluster MC1 characteristic metabolite C_10_H_8_NO_4_ was annotated as 4-(2-aminophenyl)-2,4-dioxobutanoic acid or 1-nitro-5,6-dihydroxy-dihydronaphthalene with the form of [C_10_H_9_NO_4_-H]^−^. 4-(2-aminophenyl)-2,4-dioxobutanoic acid serves as a substrate for mitochondrial kynurenine/alpha-aminoadipate aminotransferase and is involved in tryptophan metabolism, which abnormalities may exacerbate kidney injury and mediate kidney fibrosis [68,69]. Cluster MC2 characteristic metabolite C_12_H_17_N_5_O_3_SCl was annotated as 5′-ethylthioadenosine with the form of [C_12_H_17_N_5_O_3_S^+^Cl]^−^.

For the outer medulla, clusters MC3, MC4, and MC5 were prominent, with metabolites including C_6_H_11_O_6_, C_10_H_12_N_4_O_5_Cl, and C_5_H_7_O_4_ enriched in specific clusters (Appendix A). The metabolite of C_10_H_12_N_4_O_5_Cl in MC4, which was annotated to inosine or allopurinol riboside, was characterized by its chemical structure that includes the addition of a chlorine atom, denoted as [C_10_H_12_N_4_O_5_^+^Cl]^−^. These two metabolites are associated with the purine metabolic pathway, playing significant roles within the biological system and maintaining a close relationship with human health [70].

Similarly, clusters MC6 and MC7 were observed for the inner medulla, with metabolites like C_21_H_38_O_6_P and C_10_H_14_N_5_O_7_PCl enriched in specific clusters (Appendix A). The C_21_H_38_O_6_P is annotated as CPA18:1, with the formula C_21_H_39_O_6_P after deprotonation. CPA serves as a second messenger and physiological inhibitor of PPARG, participating in the regulation of adipogenesis, glucose homeostasis, and processes associated with type 2 diabetes [71]. It might be further linked to the pathogenesis and progression of DKD. Another metabolite C_10_H_14_N_5_O_7_PCl was annotated as 3′-AMP, adenosine monophosphate, or adenosine 2′-phosphate, with the form of [C_10_H_14_N_5_O_7_P^+^Cl]^−^. When cellular energy levels decrease, AMP concentrations increase, subsequently activating AMPK. AMPK serves as a key regulator of energy production in most cells and might potentially exert a protective effect against kidney injury [72,73]. 

Next, we explored the differential expressions of metabolites between LDKD and healthy samples (Figure 4f). Among these, 3-indoleacetonitrile (C_10_H_7_N_2_), sulfates (HO_4_S), organic acids like propanoic acid (C_12_H_12_NO_3_), and fatty acids such as tridecatrienoic acid (C_16_H_21_O_3_Cl_2_) and octadecanoic acid (C_18_H_36_O_3_Cl) or hydroxyoctadecanoic acid were highly expressed in LDKD (Figure 4f). In contrast, metabolites which predominantly associated with biological membranes and cellular structures were highly expressed in healthy samples, such as phospholipids (e.g., PC4:0 (C_12_H_25_NO_7_P), PE19:2 (C_24_H_43_NO_8_P)), and certain acid metabolites (e.g., 2-bromo-2-butenoic acid (C_4_H_4_O_2_Br), tetradecatetraene-8,10-diynoic acid (C_14_H_11_O_2_), and 7,8-dihydropteroic acid (C_14_H_13_N_6_O_3_)) (Figure 4f). Furthermore, antioxidants and defense molecules, such as the flavonoid compound 5,7-Dihydroxy-3′,4′-dimethoxy-5′-prenylflavanone (C_22_H_23_O_6_), were decreased in LDKD (Figure 4f). These kinds of metabolites have antioxidant properties and protect organisms from oxidative stress. Among all differentially expressed metabolites, more than 35% were lipid-related metabolites (Figure 4g). Specifically for categorizing lipid metabolites, glycerophospholipids dominated absolutely, accounting for 59% of the total, and among glycerophospholipids, PE (phosphatidylethanolamine) is an abundant differentially expressed metabolite.

### 3.5. Characterization of Lipid Distribution and Its Metabolic Implications

Lipid accumulation is recognized as a key factor contributing to disease progression in LDKD [74], mainly manifested by dysregulation of lipid oxidation, lipid uptake, and lipogenesis. We investigated the spatial distribution characteristics of lipids (Appendix A) and their correlation with spatial metabolic clusters (Figure 5a–c and Appendix A). The MC2 cluster was specific to LDKD samples. Here, triglycerides (TGs) were found to be predominantly distributed in the MC2 cluster (Figure 5a), also known as the cortical area of the kidney [75], and their elevation may increase the risk of kidney failure [76]. 

Phosphatidylethanolamine (PE) has been identified as associated with the progression of DKD [77]. As the severity of chronic kidney disease (CKD) increases, the acyl chain of PE progressively lengthens and becomes more unsaturated. Notably, the distribution of PE significantly overlaps with the MC6 and MC7 clusters (Figure 5b and Appendix A). In addition, other glycerophospholipids such as PI, PS, PG, PA, and CPA showed significant features in spatial distribution. We found that the PI and PG were co-distributed in the MC6 and MC2 clusters (Appendix A), PS was mainly concentrated in the inner medullary region in the MC6 and MC7 clusters (Appendix A), whereas PA was correlated with MC1 and MC6 clusters or MC2 and MC7 clusters (Appendix A), and CPA was highly associated with MC6 and MC7 clusters (Appendix A).

In the sphingolipid pathway, many sphingolipid-related lipids are closely associated with DKD [78]. We found that sphingomyelin (SM) and glucosylceramide (GalCer) were mainly distributed in the MC6 cluster (Figure 5c and Appendix A), which belongs to the inner medullary region. In addition, the sphingolipid pathway metabolite phosphoethanolamine (PEA), a precursor of phosphatidylcholine (PC) and PE, is predominantly distributed in the inner medullary region (Figure 5d). This distribution was highly correlated with the MC7 metabolic cluster (Figure 5e). PEA is known to attenuate inflammation, mitochondrial oxygen consumption, oxidative stress, mitochondrial fragmentation, and tubular injury in acute kidney injury [79].

## 4. Discussion

DKD is one of the most common complications among diabetic patients and a leading cause of end-stage renal disease (ESRD) [1,2,3], with its rising incidence drawing increasing attention. This study aimed to explore the metabolic profile of patients with advanced diabetic nephropathy to understand the role of lipid metabolism in the progression of kidney fibrosis. To this end, we have uncovered novel insights into the spatial transcriptome and metabolite alterations associated with LDKD by integrating spatial multi-omics data from KPMP [9]. Our findings elucidate the intricate relationship between lipid metabolism and the progression of LDKD, highlighting the potential for therapeutic interventions targeting fatty acid metabolism and the fibrosis process. Notably, our research identified spatial regions showing significant metabolite profiles in the kidney and linked to specific zonation metabolites, which were previously unrecognized. Furthermore, we discovered that the kidney cellular composition and heterogeneity are significantly altered in LDKD, with particular emphasis on the roles of injured thick ascending limb (iTAL) and injured proximal tubule (iPT) cell types. These findings enhance our understanding of LDKD and provide a scientific rationale for therapeutic strategies.

Our study focuses on identifying the characteristic features of iPT and iTAL cells. In LDKD, iTAL and iPT cell types showed increased activities related to sphingolipid and membrane lipid metabolism, as well as other lipid metabolic and biosynthetic processes, which was found in both scRNA-seq and spatial transcriptome data. This upregulation may be a stress response [80,81], with damaged cells enhancing lipid biosynthesis for membrane repair [82], maintaining fluidity, and storing energy in lipid droplets. Conversely, there was a significant decrease in their involvement in lipid and fatty acid metabolic and oxidative processes compared to their counterpart cell types, likely due to mitochondrial dysfunction [83,84]. Damaged cells often experience impaired oxidative processes [85,86] and turn to glycolysis for energy. Moreover, the iTAL cell types specifically exhibited upregulation in pathways related to steroid hormone stimulus and response. 

Here, we identified several upregulated pathway genes in iPT cells that were involved in insulin secretion from pancreatic islet β-cells. For example, FSHR is linked to postmenopausal diabetes and kidney tubulointerstitial injury [40,41]. *ABCA12* regulates pancreatic lipid homeostasis and insulin secretion [42]. Additionally, genes like *BMP7* have anti-fibrotic effects in DKD [43,44], while *CLDN1*, suppressed by *SIRT1* in kidney podocytes, mitigates diabetic albuminuria [45]. The choline transporter-like family SLC44 gene *SLC44A5* is also implicated in the increased pathways of iPT cells [61]. The significant downregulated gene *FABP1* in iPT cells increased with the progression of DKD [46], and *HSD11B2* linked to intra-adipose cortisol levels and peripheral glucose uptake [62], may be crucial for optimizing steroid therapy [63]. In iTAL cells, the typical upregulated gene *ANXA3* acts as an endogenous anti-inflammatory mediator in various diseases [47,48]. *IGFBP2* increases inflammation, and oxidative stress, and promotes podocyte apoptosis, key features of DKD [49].

Our study also indicates that sphingolipid metabolism is significantly elevated in the iTAL and iPT cell types and suggests a potential role in the pathogenesis of LDKD. Sphingolipids, as essential constituents of cellular and organelle membranes, served as key mediators in signal transduction and energy metabolism. Disruptions within the sphingolipid metabolic pathway can significantly impact the functionality and structural integrity of kidney cells [29,30]. Corroborating our observations, the literature suggested a correlation between the lipotoxicity associated with diabetic complications [9] and the accumulation of ceramides in tissue, which often preceded the onset of these conditions. This accumulation of ceramides, a subclass of sphingolipids, may elucidate the widespread upregulation observed in the sphingolipid metabolism of these clusters.

Furthermore, cholesterol metabolism in the two injury-associated cell types is significantly upregulated. This increased activity may point to enhanced cholesterol accumulation within the glomeruli, which could potentially worsen endothelial damage and lead to further dysfunction [31]. Further investigation into the cholesterol signaling metabolism network revealed that LDKD typically involved a greater number of ligand–receptor pairs. Notably, *RORA* and *RORC* are implicated in this process. As members of the nuclear receptor family of intracellular transcription factors, both of these receptor molecules are associated with fibrotic processes [32,33,64,65]. Meanwhile, the sender gene *DHCR24* is involved in cholesterol biosynthesis and acts as an antioxidant [34,35]. It also has proapoptotic functions and may serve as a novel biomarker for the risk of type 2 diabetes [36,37]. In a spatial context, cholesterol exhibits more complex cell–cell interactions. 

Additionally, spatial transcriptomics analysis revealed that genes exhibit significant functional dependencies on their spatial positioning within the kidney. This allows us to further explore the transcriptional landscape in LDKD patients and identify key genes that show significant changes between injury-associated clusters and their corresponding clusters in healthy samples. In LDKD, the genes commonly increased in the two injury-associated cell types are primarily related to injury and fibrosis. For instance, *SERPINA3* is a biomarker for renal tubular injury in DKD [50]. *VCAM1* indicates tubular injury and predicts poor outcomes in DKD [51,52]. *ITGB6* is associated with fibrosis and injury processes [18]. *LTF*, an innate immune protein, has anti-inflammatory properties [53]. The genes increase in iTAL and decrease in iPT cells occur early during DKD. *GPX3* is linked to fat accumulation [54,55] and early kidney damage [56,57]. *SLC12A3* contributes to genetic susceptibility to DKD [58,59], with its polymorphisms associated with end-stage renal disease [60]. This suggests that iTAL is associated with the early stage of LDKD. 

Next, spatial metabolomics data analysis also provided insights into the spatial metabolite distribution characteristic in LDKD. We identified regional metabolic clusters specific to LDKD patients and observed that lipid-associated metabolites, particularly glycerophospholipids, represented a significant portion of the differentially expressed metabolites between LDKD patients and healthy participants. Metabolites associated with sphingolipid metabolism, including triglycerides (TG), glycerophospholipids, and sphingomyelin (SM), were notably concentrated in the inner medullary regions. TG and PE increase the risk of kidney failure [76] and are associated with the progression of DKD [77]. The sphingolipid pathway metabolite phosphoethanolamine (PEA) is known to attenuate inflammation and tubular injury in acute kidney injury [79]. This deepens our understanding of lipid-related pathways in LDKD and healthy conditions from a metabolic perspective.

Then, we would like to discuss the relationship between lipid metabolism and fibrosis. In specific kidney regions of LDKD patients, especially the inner medullary area, we observed a high concentration of metabolites related to lipid metabolism, such as glycerophospholipids, TG, PE, and sphingolipids. This accumulation is linked to an increased risk of renal failure and DKD progression [77]. Disrupted sphingolipid metabolism, which plays a critical role in cell signaling and energy regulation, contributes to renal cell injury and fibrosis. Notably, phosphoethanolamine (PEA) has been shown to reduce inflammation and tubular injury, suggesting a role in mitigating fibrosis [79]. Additionally, cholesterol metabolism abnormalities exacerbate endothelial damage and fibrosis, with receptors like *RORA* and *RORC*, playing key roles in these processes [32,33,64,65]. Overall, changes in sphingolipid and cholesterol metabolism are closely tied to renal fibrosis and may offer targets for treating diabetic nephropathy.

However, certain limitations should be acknowledged. A primary limitation of this study is the small sample size, which may affect the generalizability of these findings. Additionally, the current spatial omics approach has technical limitations. Firstly, aligning spatial spots directly between different spatial technologies presents difficulties due to differences in spatial spot sampling methods and variability across tissue sections. These issues will be addressed in our future research to enhance data alignment and comparability. Secondly, MALDI mass spectrometry, optimized for larger molecules like phospholipids and sphingolipids [87], faces challenges with smaller, less polar metabolites due to ionization and matrix interference, especially in complex tissue samples. The matrix, essential for ionization, can produce background signals that mask short-chain fatty acids, and the lack of polar groups reduces ionization efficiency, impacting sensitivity. Furthermore, fatty acid isomers complicate accurate identification, requiring secondary mass spectrometry or lipid-specific techniques like LC-MS or GC-MS.

Despite these challenges, our study reveals a significant association between lipid metabolism and fibrosis in LDKD patients through spatial multi-omics analysis, highlighting the roles of phospholipids and sphingolipids in damaged kidney regions. It reveals distinct metabolic characteristics of LDKD by identifying unique spatial metabolic signatures and highlighting specific lipid metabolic pathways as potential therapeutic targets. By identifying unique metabolic clusters, our study connects sphingolipid and cholesterol metabolism with fibrosis and kidney injury, forming a foundation for targeted therapeutic interventions that may improve LDKD management and patient outcomes. This integrative multi-omics study not only deepens the understanding of LDKD’s metabolic dysregulation but also offers new avenues for future research, paving the way for precision medicine in LDKD treatment and management.

## 5. Conclusions

The study provides an integrative view of lipid metabolism in long-standing diabetic kidney disease, highlighting dysregulated metabolic pathways and regionally distributed lipid metabolites. These findings enhance understanding of injury processes in diabetic kidney disease and their role in fibrosis. 

## Figures and Tables

**Figure 1 metabolites-14-00641-f001:**
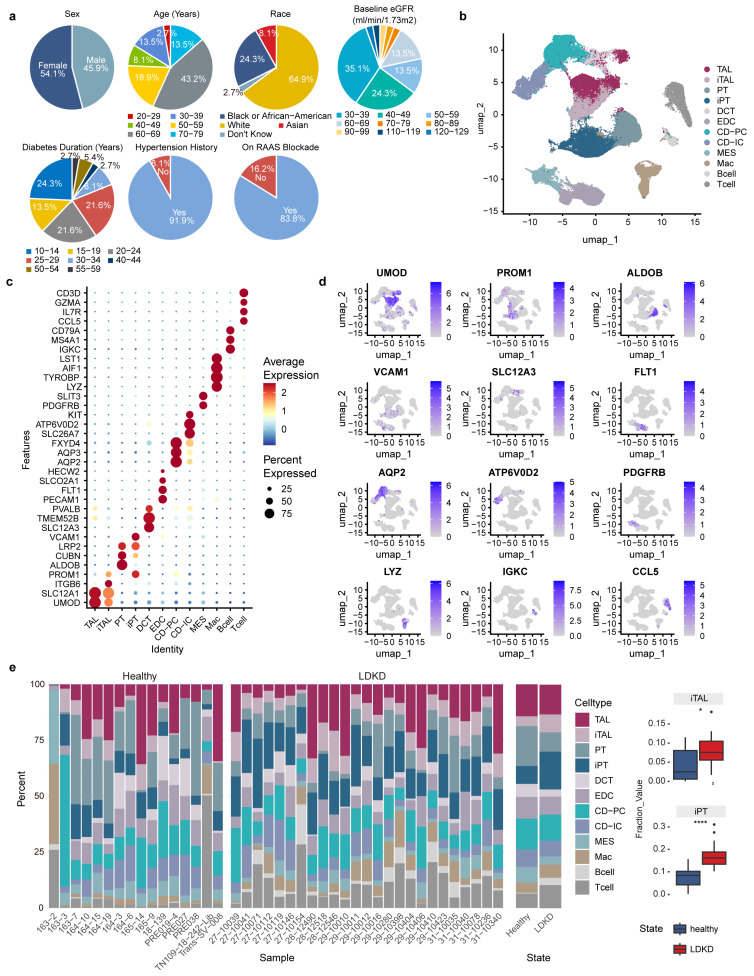
Single-cell RNA sequencing (scRNA-seq) analysis reveals kidney cell types in LDKD and healthy donors. (**a**) Clinical profiles of LDKD patients in the study. RAAS, renin–angiotensin–aldosterone system. (**b**) UMAP diagram of the identified cell types. Different colors correspond to distinct cell types. (**c**) Dot plot of the markers corresponding to the cell types. (**d**) UMAP diagram of the expression of canonical markers for the cell types. The color scales across multiple plots were adjusted by gene scaling. (**e**) Bar plot of the composition of different cell types in each sample. Alongside are the proportions of iTAL and iPT cell types in LDKD and healthy samples. Wilcoxon test. * *p* ≤ 0.05, **** *p* ≤ 0.0001.

**Figure 2 metabolites-14-00641-f002:**
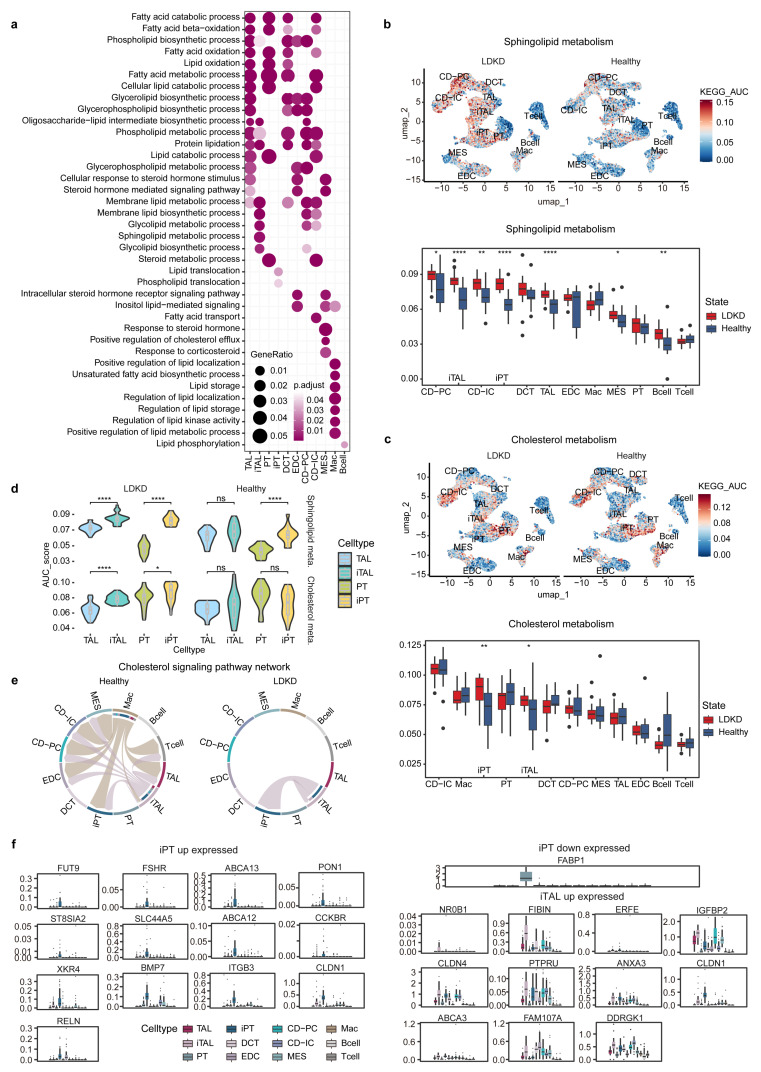
Metabolic pathway profiles of different kidney cell types based on the scRNA-seq data. (**a**) Dot plot showing lipid-associated metabolism process enrichment (from GO BP) across cell types. (**b**,**c**) UMAP plots of sphingolipid (**b**) and cholesterol (**c**) metabolism pathway activities in LDKD vs. healthy samples. Box plots compare pathway activities between LDKD and healthy samples by cell type (Wilcoxon test: * *p* ≤ 0.05, ** *p* ≤ 0.01, **** *p* ≤ 0.0001, ns: not significant). The black points are outliers in each boxplot. (**d**) Violin plots comparing sphingolipid and cholesterol metabolism pathway activities between iTAL/TAL and iPT/PT in LDKD (left) vs. healthy (right). Wilcoxon test. (**e**) Chord diagram of cholesterol metabolism pathway in cell–cell interaction networks for healthy and LDKD samples. (**f**) Differentially expressed genes in iPT (up/downregulated) and iTAL (upregulated) pathways.

**Figure 3 metabolites-14-00641-f003:**
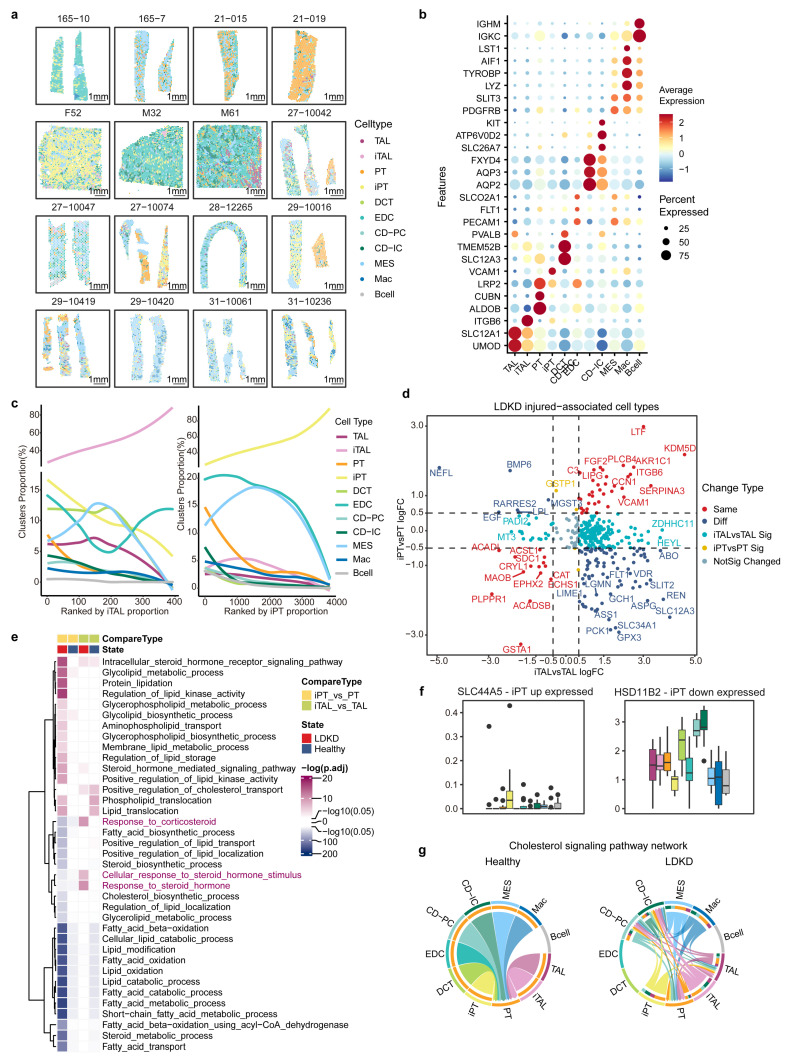
Spatial transcriptomics analysis for LDKD and healthy samples. (**a**) Spatial distribution of cell types in LDKD and healthy samples. Scale bar: 1 mm. (**b**) Dot plot showing characteristic markers for distinct cell types. (**c**) Loess smoothed curves of cell type proportion changes with iTAL (**left**) and iPT (**right**). The x-axis ranks cells by iTAL/iPT proportion; the lower half of the y-axis shows changes in other cell types’ proportions. (**d**) Scatter plot of differentially expressed genes in iTAL vs. TAL and iPT vs. PT comparisons in LDKD samples. Colors indicate avg_logFC direction, with a 0.5 threshold for separation. Wilcoxon test. (**e**) Pathway activities comparison between iTAL vs. TAL and iPT vs. PT in LDKD and healthy samples. Dark pink/blue indicates positive/negative activity with intensity reflecting adjusted *p*-value. (**f**) Differentially expressed genes in iPT up/downregulated pathways. Genes show significant differences (*p* < 0.05) comparing injured cells to other types and counterparts. The black points are outliers in each boxplot. Different colors represent different cell types. (**g**) Chord plot of cholesterol metabolism pathway in cell–cell interactions for healthy and LDKD samples, with cell types exceeding 50% included.

**Figure 4 metabolites-14-00641-f004:**
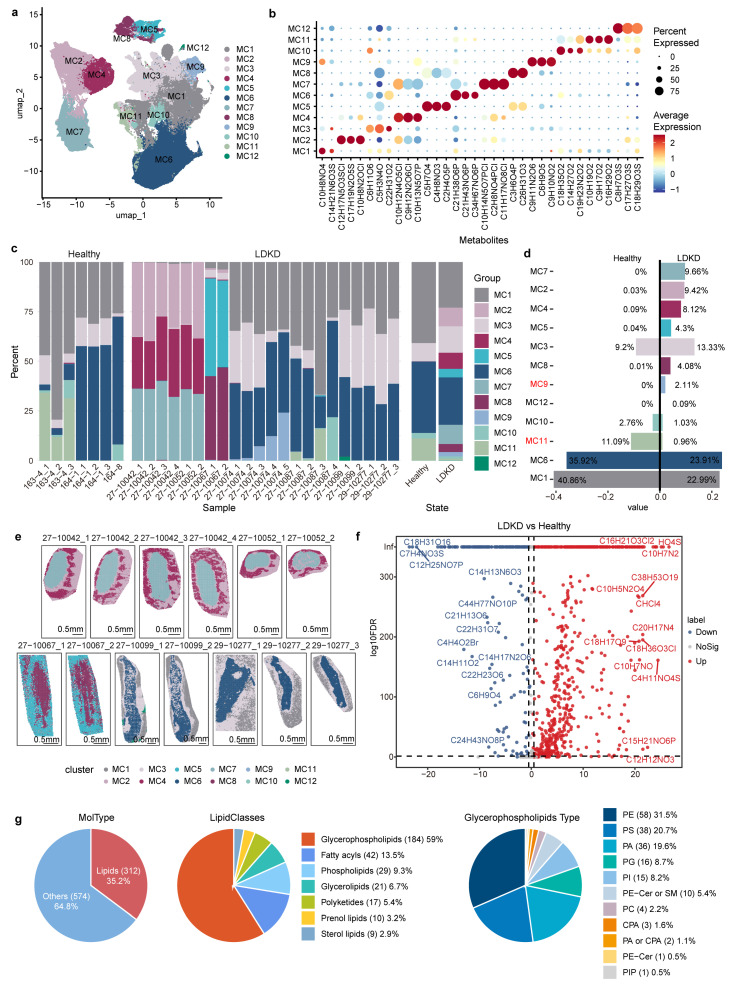
Spatial metabolomics analysis for LDKD and healthy samples. (**a**) UMAP plot of spatial metabolic clusters (MCs). Different colors correspond to distinct clusters. (**b**) Dot plot of characteristic metabolites in different MCs. (**c**) Bar plot of the composition of MCs in each sample. (**d**) Pyramid chart of MC proportion changes between LDKD and healthy samples, ordered by absolute LDKD proportion change, with significantly changed clusters in red. (**e**) Spatial distribution of MCs in LDKD samples, with colors consistent with A. Scale bar: 0.5 mm. (**f**) Differentially expressed metabolites between LDKD and healthy samples identified via Wilcoxon test, with Bonferroni-adjusted *p*-values. (**g**) Pie chart showing proportions of different metabolite/lipid/glycerophospholipid classes among differentially expressed metabolites. Each category is colored distinctly, with the name followed by the number of matches and percentage.

**Figure 5 metabolites-14-00641-f005:**
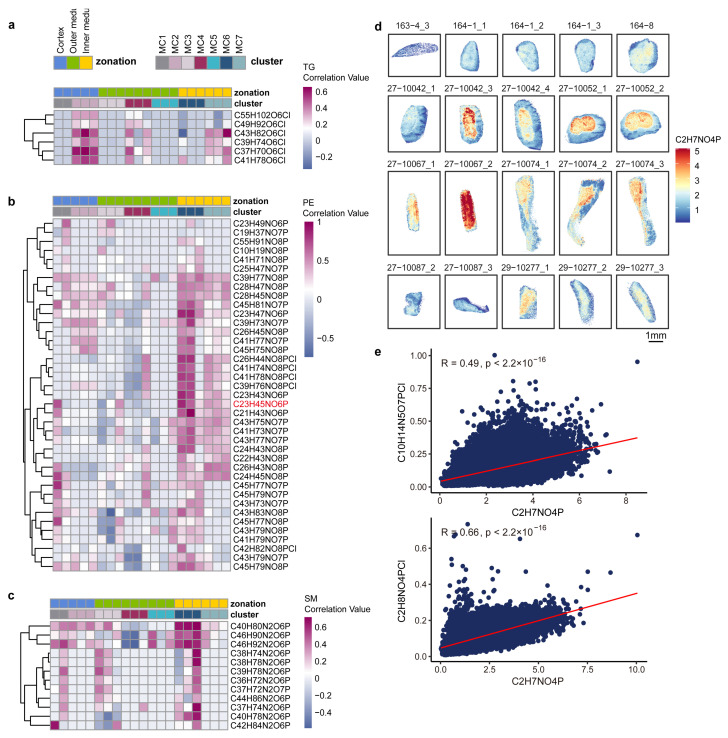
Spatial distribution similarities between lipids and characteristic metabolites of different MCs in LDKD. (**a**–**c**) Spatial distribution similarity of TG (**a**), PE (**b**), and SM (**c**) with metabolic clusters in LDKD. The heatmap colors represent the correlation between each metabolite and different metabolic clusters, defined by their spatial distribution similarity. Kidney regions and metabolic clusters are distinguished by distinct colors and annotations. TG, triglycerides; PE, phosphatidylethanolamine; SM, sphingomyelin. Metabolites labelled in red are shown in figS6. For each MC, 2 or 3 characteristic metabolites were selected to characterize the spatial distribution. (**d**) Spatial distribution of the metabolite C_2_H_7_NO_4_P, which annotated PEA. Scale bar, 1 mm. PEA, phosphoethanolamine. (**e**) Spatial distribution similarity of metabolite C_2_H_7_NO_4_P with MC7 characteristic metabolites C_10_H_14_N_5_O_7_PCl and C_2_H_8_NO_4_PCl. The red line in each plot is the regression line for all points.

## Data Availability

The scRNA-seq and spatial multi-omics data used in the study are available from the Kidney Precision Medicine Project (KPMP) at https://www.kpmp.org/, accessed on 7 October 2023.

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
