# Peer review of "Identification of Spatial Specific Lipid Metabolic Signatures in Long-Standing Diabetic Kidney Disease"

_metabolites, 2024, doi:10.3390/metabo14110641_

Round 1
Reviewer 1 Report
Comments and Suggestions for Authors
This paper identifies spatial specific lipid metabolic signatures in long-term diabetic nephropathy by integrative analyses of single-cell transcriptomics, spatial transcriptomics, and spatial metabolomics data from Kidney Precision Medicine Project cohort and delineated the metabolic signatures associated with advanced diabetes, thereby providing insights into the disease process. My observations/remarks:
1) Line 41: Membranes are actually phospholipid bilayers, but in this line, author mentions ‘phospholipids, and cellular membrane lipids…’. What are these other membrane lipids as phospholipids and sterols have already been mentioned. May please specify. Sphingomyelin?
2) It has been over a year when the authors retrieved the data. A year is a quite long duration.
3) Diabetic history is important as is the treatment and treatment method. Author may like to elaborate exclusion/inclusion criteria.
4) How the specific n values were arrived at, for example, n=10 in spatial metabolomics, n=16 in spatial transcriptomics, and n=41 in single-cell transcriptomics?
5) The abbreviation LDKD was first used in line 54, but its expanded form appears in line 175.
6) What is the correlation of an increased lipid metabolic/biosynthetic activity and decreased lipid/fatty acid oxidative processes in injured TAL/PT cells compared to normal. Does not oxidative process be more in injured cells?
7) Authors may like to elaborate in discussion section relationship b/w lipid metabolism and fibrosis.
8) Language of the manuscript is OK, but at places, sentences have been made complex; for example, line 27-29 could have been simply written as: ‘Diabetic kidney disease (DKD) is a predominant cause of end-stage renal disease (ESRD) … prevalence closely linked to lifestyle and dietary habits’, instead of using words like ‘modern evolution of lifestyle’ etc.
9) Authors may like to add a small paragraph on the limitation of their study.
Comments on the Quality of English LanguageFine, but may be improved.
Author Response
Reviewer 1
This paper identifies spatial specific lipid metabolic signatures in long-term diabetic nephropathy by integrative analyses of single-cell transcriptomics, spatial transcriptomics, and spatial metabolomics data from Kidney Precision Medicine Project cohort and delineated the metabolic signatures associated with advanced diabetes, thereby providing insights into the disease process. My observations/remarks:
1) Line 41: Membranes are actually phospholipid bilayers, but in this line, author mentions ‘phospholipids, and cellular membrane lipids…’. What are these other membrane lipids as phospholipids and sterols have already been mentioned. May please specify. Sphingomyelin?
Reply: Thanks for the correction, what we were trying to convey was indeed sphingomyelin, and we have revised this.
2) It has been over a year when the authors retrieved the data. A year is a quite long duration.
Reply: Thank you for your question. A year is indeed a long time, and we are analyzing data on a MAFLD-related article in parallel with the analysis. The MAFLD-related article was accepted by Life Metabolism on 2024.05.31 (https://doi.org/10.1093/lifemeta/loae021).
3) Diabetic history is important as is the treatment and treatment method. Author may like to elaborate exclusion/inclusion criteria.
Reply: Thank you for your question.
The treatment of LDKD patients was taken into account, specifically in the ‘On RAAS Blockade’ metric, and there is a lot of literature showing that blockade of the renin-angiotensin-aldosterone system (RAAS) by blocking agents has antihypertensive and antiproteinuric properties, and is a cornerstone of diabetic nephropathy treatment [1-3]. Thirty-one of the 37 LDKD patients involved in all analyses underwent RAAS blocker therapy. However, the treatment method detail in KPMP cohort is not clear.
To select the samples for the study, we first screened health and chronic kidney disease (CKD) patients. To elucidate the metabolic characteristics of LDKD patients with prolonged diabetes duration, we focused on individuals with over a decade of diabetes history. For the spatial multi-omics sliced samples, which additionally involved the spatial distributions, we removed samples with spatial discontinuities. A total of 37 LDKD patients and 26 healthy donors were included in the study.
The detailed information about these collected samples including the “On RAAS Blockade” was recorded in Table S1.
4) How the specific n values were arrived at, for example, n=10 in spatial metabolomics, n=16 in spatial transcriptomics, and n=41 in single-cell transcriptomics?
Reply: Thank you for your question.
We provided a more detailed explanation of the sample collection process as outlined below:
To select the samples for the study, we first screened health and chronic kidney disease (CKD) patients. To elucidate the metabolic characteristics of LDKD patients with prolonged diabetes duration, we focused on individuals with over a decade of diabetes history. For the spatial multi-omics sliced samples, which additionally involved the spatial distributions, we removed samples with spatial discontinuities. A total of 37 LDKD patients and 26 healthy donors were included in the study. The scRNA-seq data included samples from 25 LDKD patients and 16 healthy donors. The spatial transcriptomics data included samples from 9 LDKD and 7 healthy donors. The spatial metabolomics data included samples from 7 LDKD and 3 healthy donors. The detailed information for the selected samples was summarized in Table S1.
5) The abbreviation LDKD was first used in line 54, but its expanded form appears in line 175.
Reply: Thanks for the suggestion, we have added its extended definition where it first appeared.
6) What is the correlation of an increased lipid metabolic/biosynthetic activity and decreased lipid/fatty acid oxidative processes in injured TAL/PT cells compared to normal. Does not oxidative process be more in injured cells?
Reply: Thank you for your question and we have added what you mentioned in the discussion section.
This increased lipid metabolic/biosynthetic activity in injured TAL/PT cells may be a stress response [4,5], with damaged cells enhancing lipid biosynthesis for membrane repair [6], maintaining fluidity, and storing energy in lipid droplets.
The decreased lipid/fatty acid oxidative processes in injured TAL/PT cells are likely due to mitochondrial dysfunction [7,8]. Damaged cells often experience impaired oxidative processes [9,10] and turn to glycolysis for energy.
Oxidative stress often exacerbates damage. Cells may limit fatty acid oxidation to reduce harmful reactive oxygen species (ROS) production. This shift between lipid biosynthesis and fatty acid oxidation represents a protective response to injury, prioritizing cell survival over efficient energy production.
7) Authors may like to elaborate in discussion section relationship b/w lipid metabolism and fibrosis.
Reply: Thank you for your question, we have added what you mentioned in the discussion section.
There is a strong link between lipid metabolism and renal fibrosis.
In LDKD patients’ specific kidney regions, especially the inner medullary area, showed a high concentration of metabolites related to lipid metabolism, such as glycerophospholipids, TG, PE, and sphingolipids. This accumulation is linked to an increased risk of renal failure and DKD progression [11]. Disrupted sphingolipid metabolism, which is vital for cell signaling and energy regulation, contributes to renal cell injury and fibrosis. Notably, phosphoethanolamine (PEA) has been shown to reduce inflammation and tubular injury, suggesting a role in mitigating fibrosis [12]. Additionally, cholesterol metabolism abnormalities exacerbate endothelial damage and fibrosis, with receptors like RORA and RORC, playing key roles in these processes [13-16]. Overall, changes in sphingolipid and cholesterol metabolism are closely tied to renal fibrosis and may offer targets for treating diabetic nephropathy.
8) Language of the manuscript is OK, but at places, sentences have been made complex; for example, line 27-29 could have been simply written as: ‘Diabetic kidney disease (DKD) is a predominant cause of end-stage renal disease (ESRD) … prevalence closely linked to lifestyle and dietary habits’, instead of using words like ‘modern evolution of lifestyle’ etc.
Reply: Thank you for your suggestions, we have made improvement.
9) Authors may like to add a small paragraph on the limitation of their study.
Reply: Thank you for your question and we have added what you mentioned in the discussion section.
However, certain limitations should be acknowledged. A primary limitation of this study is the small sample size, which may affect the generalizability of these findings. Additionally, the current spatial omics approach has technical limitations. Firstly, aligning spatial spots directly between different spatial technologies presents difficulties due to differences in spatial spot sampling methods and variability across tissue sections. These issues will be addressed in our future research to enhance data alignment and comparability. Secondly, MALDI mass spectrometry, optimized for larger molecules like phospholipids and sphingolipids [23], faces challenges with smaller, less polar metabolites due to ionization and matrix interference, especially in complex tissue samples. The matrix, essential for ionization, can produce background signals that mask short-chain fatty acids, and the lack of polar groups reduces ionization efficiency, impacting sensitivity. Besides, fatty acid isomers complicate accurate identification, requiring secondary mass spectrometry or lipid-specific techniques like LC-MS or GC-MS.
References
- Lin, Y.C.; Chang, Y.H.; Yang, S.Y.; Wu, K.D.; Chu, T.S. Update of pathophysiology and management of diabetic kidney disease. J Formos Med Assoc 2018, 117, 662-675, doi:10.1016/j.jfma.2018.02.007.
- Roscioni, S.S.; Heerspink, H.J.L.; de Zeeuw, D. The effect of RAAS blockade on the progression of diabetic nephropathy. Nat Rev Nephrol 2014, 10, 77-87, doi:10.1038/nrneph.2013.251.
- Sawaf, H.; Thomas, G.; Taliercio, J.J.; Nakhoul, G.; Vachharajani, T.J.; Mehdi, A. Therapeutic Advances in Diabetic Nephropathy. J Clin Med 2022, 11, doi:10.3390/jcm11020378.
- Asowata, E.O.; Romoli, S.; Sargeant, R.; Tan, J.Y.; Hoffmann, S.; Huang, M.M.; Mahbubani, K.T.; Krause, F.N.; Jachimowicz, D.; Agren, R.; et al. Multi-omics and imaging mass cytometry characterization of human kidneys to identify pathways and phenotypes associated with impaired kidney function. Kidney International 2024, 106, 85-97, doi:10.1016/j.kint.2024.01.041.
- Feng, Y.; Sun, Z.G.; Fu, J.; Zhong, F.; Zhang, W.J.; Wei, C.G.; Chen, A.Q.; Liu, B.C.; He, J.C.; Lee, K. Podocyte-derived soluble RARRES1 drives kidney disease progression through direct podocyte and proximal tubular injury. Kidney International 2024, 106, 50-66, doi:10.1016/j.kint.2024.04.011.
- Zuo, S.M.; Wang, Y.X.; Bao, H.J.; Zhang, Z.H.; Yang, N.F.; Jia, M.; Zhang, Q.; Jian, A.N.; Ji, R.; Zhang, L.D.; et al. Lipid synthesis, triggered by PPARγ T166 dephosphorylation, sustains reparative function of macrophages during tissue repair. Nat Commun 2024, 15, doi:10.1038/s41467-024-51736-5.
- Bhargava, P.; Schnellmann, R.G. Mitochondrial energetics in the kidney. Nat Rev Nephrol 2017, 13, 629-646, doi:10.1038/nrneph.2017.107.
- Narongkiatikhun, P.; Choi, Y.J.; Hampson, H.; Gotzamanis, J.; Zhang, G.; van Raalte, D.H.; de Boer, I.H.; Nelson, R.G.; Tommerdahl, K.L.; McCown, P.J.; et al. Unraveling Diabetic Kidney Disease: The Roles of Mitochondrial Dysfunction and Immunometabolism. Kidney Int Rep, doi:10.1016/j.ekir.2024.09.019.
- Faivre, A.; Verissimo, T.; Auwerx, H.; Legouis, D.; de Seigneux, S. Tubular Cell Glucose Metabolism Shift During Acute and Chronic Injuries. Front Med-Lausanne 2021, 8, doi:10.3389/fmed.2021.742072.
- Kang, H.M.; Ahn, S.H.; Choi, P.; Ko, Y.A.; Han, S.H.; Chinga, F.; Park, A.S.D.; Tao, J.L.; Sharma, K.; Pullman, J.; et al. Defective fatty acid oxidation in renal tubular epithelial cells has a key role in kidney fibrosis development. Nat Med 2015, 21, 37-46, doi:10.1038/nm.3762.
- Afshinnia, F.; Nair, V.; Lin, J.; Rajendiran, T.M.; Soni, T.; Byun, J.; Sharma, K.; Fort, P.E.; Gardner, T.W.; Looker, H.C.; et al. Increased lipogenesis and impaired β-oxidation predict type 2 diabetic kidney disease progression in American Indians. Jci Insight 2019, 4, e130317, doi:10.1172/jci.insight.130317.
- Kishi, S.; Campanholle, G.; Gohil, V.M.; Perocchi, F.; Brooks, C.R.; Morizane, R.; Sabbisetti, V.; Ichimura, T.; Mootha, V.K.; Bonventre, J.V. Meclizine Preconditioning Protects the Kidney Against Ischemia-Reperfusion Injury. Ebiomedicine 2015, 2, 1090-1101, doi:10.1016/j.ebiom.2015.07.035.
- Dikun, K.M.; Tang, X.H.; Fu, L.P.; Choi, M.E.; Lu, C.Y.; Gudas, L.J. Retinoic acid receptor α activity in proximal tubules prevents kidney injury and fibrosis. P Natl Acad Sci USA 2024, 121, e2311803121, doi:10.1073/pnas.2311803121.
- Zhong, Y.F.; Wu, Y.W.; Liu, R.J.; Li, Z.Z.; Chen, Y.B.; Evans, T.; Chuang, P.; Das, B.; He, J.C. Novel Retinoic Acid Receptor Alpha Agonists for Treatment of Kidney Disease. Plos One 2011, 6, e27945, doi:10.1371/journal.pone.0027945.
- Jetten, A.M.; Cook, D.N. (Inverse) Agonists of Retinoic Acid-Related Orphan Receptor γ: Regulation of Immune Responses, Inflammation, and Autoimmune Disease. Annu Rev Pharmacol 2020, 60, 371-390, doi:10.1146/annurev-pharmtox-010919-023711.
- Hu, X.; Wang, Y.H.; Hao, L.Y.; Liu, X.K.; Lesch, C.A.; Sanchez, B.M.; Wendling, J.M.; Morgan, R.W.; Aicher, T.D.; Carter, L.L.; et al. Sterol metabolism controls T(H)17 differentiation by generating endogenous RORγ agonists. Nat Chem Biol 2015, 11, 141-147, doi:10.1038/Nchembio.1714.

Reviewer 2 Report
Comments and Suggestions for Authors
This manuscript profiling the spatial lipid metabolic signatures in long-standing diabetic kidney disease using integrative single cell RNA sequencing and spatial multi-omics. Overall, the manuscript is well written and logically clear, and is also an important report for diabetic kidney disease. I think it is suitable for publication in Metabolites. Several issues require further clarification by the authors.
1. The author analyzed spatial metabolomics data, but why did they not display the spatial distribution images of representative lipids such as PE, PC, PI, PS, PG, PA, SM, FA.
2. The identification of different lipids needs clarification.
3. Can the authors add spatial correlation analysis of lipids and key genes, which I think is very important information that spatial multi-omics can provide.
4. Fatty acid is an important class of lipids. Spatial transcriptomics analysis suggests alterations in genes associated with fatty acid metabolism. However, the spatial metabolomics data do not seem to show any significant alterations. Does the author pay attention to changes in fatty acids.
Author Response
Reviewer 2
This manuscript profiling the spatial lipid metabolic signatures in long-standing diabetic kidney disease using integrative single cell RNA sequencing and spatial multi-omics. Overall, the manuscript is well written and logically clear, and is also an important report for diabetic kidney disease. I think it is suitable for publication in Metabolites. Several issues require further clarification by the authors.
1. The author analyzed spatial metabolomics data, but why did they not display the spatial distribution images of representative lipids such as PE, PC, PI, PS, PG, PA, SM, FA.
Reply: Thank you for your question. We have added Fig. S6 for the spatial distribution of the characterized metabolites of PE, PI, PS, and PA.
2. The identification of different lipids needs clarification.
Reply: Thank you for your question.
The molecular annotations and pixel intensity matrices for all samples at each sampling point in the spatial metabolomics experiment were obtained from METASPACE. The categorization information of lipid-related metabolites was obtained from the KEGG compound database and categorized by merging the lipid category information from br08001 and br08002. This is mentioned in 2.9. Spatial metabolomics data analysis and 2.10. Analysis of differential metabolite class composition of Methods.
3. Can the authors add spatial correlation analysis of lipids and key genes, which I think is very important information that spatial multi-omics can provide.
Reply: Thank you for your question, which is indeed a very important one.
There are some challenges with the current technology. First, the dataset includes spatial transcriptomics and spatial metabolomics data for only two patients, which restricts sample diversity. Second, aligning sampling points directly between spatial technologies presents difficulties due to sampling method differences and variability across tissue sections. These issues will be further addressed in future research. We have added some discussions about the issue.
4. Fatty acid is an important class of lipids. Spatial transcriptomics analysis suggests alterations in genes associated with fatty acid metabolism. However, the spatial metabolomics data do not seem to show any significant alterations. Does the author pay attention to changes in fatty acids.
Reply: Thank you for your question.
MALDI mass spectrometry, optimized for larger molecules like phospholipids and sphingolipids [1], faces challenges with smaller, less polar metabolites due to ionization and matrix interference, especially in complex tissue samples. The matrix, essential for ionization, can produce background signals that mask short-chain fatty acids, and the lack of polar groups reduces ionization efficiency, impacting sensitivity. Additionally, fatty acid isomers complicate accurate identification, requiring secondary mass spectrometry or lipid-specific techniques like LC-MS or GC-MS.
References
- Moreno-Gordaliza, E.; Esteban-Fernández, D.; Lázaro, A.; Humanes, B.; Aboulmagd, S.; Tejedor, A.; Linscheid, M.W.; Gómez-Gómez, M.M. MALDI-LTQ-Orbitrap mass spectrometry imaging for lipidomic analysis in kidney under cisplatin chemotherapy. Talanta 2017, 164, 16-26, doi:10.1016/j.talanta.2016.11.026.

Reviewer 3 Report
Comments and Suggestions for Authors
The current manuscript aims to contribute to the knowledge associated to metabolic mechanisms associated to the pathophysiological state of long-standing diabetic kidney diseases compared to health volunteers, based on sc-RNA sequencing and spatial transcriptomics and spatial metabolomics.
In general, the manuscript is well-structured and uses an objective language and a rational deduction of data interpretation. Despite this, I have the following comments:
- It is not clear on the Materials and Methods section, the exact number of volunteers used for the spatial transcriptomic and metabolomic analysis
- It is not clear, why Fig. 1 is representing the average characteristics of the 63 individuals, instead of the 41 used in the sc-RNA analysis.
- It is not clear, on the Materials and Methods Section, and on the Results and Discussion Section, why from the 41 individuals used for sc-RNA sequencing, only a sub-set of 16 individuals were used for spatial transcriptomics. Why not using the same individuals for both studies? And are this sub-set of 16 individuals from the 41 set?
- Is not clear, how metabolomics data were obtained. In the case of the transcriptomics, e.g., it is presented that were from cs-RNA sequencing. Also, why only a group of 10 individuals were used. It is not clear if the 10 individuals used for the spatial metabolomics were from the sub-set of individuals used for the spatial transcriptomics.
- It is not clear from the 16 and 10 individuals used for the spatial transcriptomic and metabolomic, what was the percentage of them with LDKD.
- Section 3.4 points that the study was directed to “different diseases state”, however only data from healthy and LDKD were used.
TThe clarification of some of these doubts can have major implications on data analysis and biological interpretation.
All this should be clear on the abstract, since will impact the outputs generalizations.
Author Response
Reviewer 3
The current manuscript aims to contribute to the knowledge associated to metabolic mechanisms associated to the pathophysiological state of long-standing diabetic kidney diseases compared to health volunteers, based on sc-RNA sequencing and spatial transcriptomics and spatial metabolomics.
In general, the manuscript is well-structured and uses an objective language and a rational deduction of data interpretation. Despite this, I have the following comments:
1. It is not clear on the Materials and Methods section, the exact number of volunteers used for the spatial transcriptomic and metabolomic analysis
Reply: Thank you for your question. We have added it in section 2.1. Participant selection and data acquisition.
2. It is not clear, why Fig. 1 is representing the average characteristics of the 63 individuals, instead of the 41 used in the sc-RNA analysis.
Reply: Thank you for your question. The main reason is that fig1A characterizes all experiments (scRNA, spatial transcriptomics, spatial metabolomics) involving healthy controls and LDKD patients, and is not limited to scRNA experiments. We added Table S1 to show the LDKD patients involved in the scRNA, spatial transcriptomics and spatial metabolomics experiments.
3. It is not clear, on the Materials and Methods Section, and on the Results and Discussion Section, why from the 41 individuals used for sc-RNA sequencing, only a sub-set of 16 individuals were used for spatial transcriptomics. Why not using the same individuals for both studies? And are this sub-set of 16 individuals from the 41 set?
Reply: Thank you for your question.
All these data were downloaded from the Kidney Precision Medicine Project (KPMP) (https://www.kpmp.org/). Firstly, we tried to find samples with both scRNA and spatial transcriptomics in the pre-analysis, and there were only 2 such patients (29-10016, 31-10236). To account for sample size, we included more samples in each experiment.
Secondly, this subset of 16 individuals is a subset of 41 individuals.
4. Is not clear, how metabolomics data were obtained. In the case of the transcriptomics, e.g., it is presented that were from sc-RNA sequencing. Also, why only a group of 10 individuals were used. It is not clear if the 10 individuals used for the spatial metabolomics were from the sub-set of individuals used for the spatial transcriptomics.
Reply: Thank you for your question.
Firstly, all experiments (scRNA, spatial transcriptomics, spatial metabolomics) were obtained from the KPMP dataset. For the samples selection of the study, we first screened health and chronic kidney disease (CKD) patients. To elucidate the metabolic characteristics of LDKD patients with prolonged diabetes duration, we focused on individuals with over a decade of diabetes history. For the spatial multi-omics sliced samples, which additionally involved the spatial distributions, we removed samples with spatial discontinuities. A total of 37 LDKD patients and 26 healthy donors were included in the study.
Secondly, there was an intersection of patients in the spatial metabolomics and spatial transcriptomics, including 27-10042 and 27-10074, both subsets of the total study population of 41 individuals. The total study population of the spatial metabolomics group included three healthy controls and seven LDKD patients, most of whom had multiple tissue samples taken, e.g., 27-10042_1 represents the 1st tissue sample from patient 27-10042, 27-10042_2 represents the 2nd tissue sample from patient 27-10042, while 164-8 would represent only patient 164-8's tissue sample.
The detailed information about the source patients of each collected sample was summarized in Table S1.
5. It is not clear from the 16 and 10 individuals used for the spatial transcriptomic and metabolomic, what was the percentage of them with LDKD.
Reply: Thank you for your question.
We have explained this in detail in section 2.1. Participant selection and data acquisition. There were 9 LDKD patients out of 16 subjects in spatial transcriptomics and 7 LDKD patients out of 10 subjects in spatial metabolomics.
6. Section 3.4 points that the study was directed to “different diseases state”, however only data from healthy and LDKD were used.
Reply: Thank you for your question. We have changed the title from ‘in different disease states’ to ‘in LDKD and healthy kidneys’ for the study.
7. The clarification of some of these doubts can have major implications on data analysis and biological interpretation. All this should be clear on the abstract, since will impact the outputs generalizations.
Reply: Thank you for your suggestion. We have revised the abstract and provided more details in the methods.
Round 2
Reviewer 2 Report
Comments and Suggestions for Authors
The author answered my previous questions and made revisions to the manuscript, which I think can be published on Meatbolites.